# The Takeover of Science Communication: How Science Lost Its Leading Role in the Public Discourse on Carbon Capture and Storage Research in Daily Newspapers in Germany

Simon Schneider[1]

[1]Institute for Geosciences, University of Potsdam, Potsdam, 14476, Germany

*Correspondence to*: Simon Schneider (simschne@uni-potsdam.de)

**Abstract.** CCS (Carbon Capture and Storage) is an important issue within the context of climate-change mitigation options and has played a major role in the agendas of scientists, researchers, and engineers. While media representations of CCS in Germany from 2004 to 2014 demonstrated the significant medialization of the topic, this cannot be ascribed to science. Instead, CCS media coverage in Germany has been dominated by other stakeholder groups. While CCS is linked to various industry sectors, such as cement and steel production, the German debate has dominantly focused on the coal and energy branches. This study looks at the role of science and science PR within the German public debate by analyzing the media coverage of CCS in daily newspapers from 2004 to 2014. If science wishes to remain proactive within science communication, new approaches for future science public relations (PR) have to be deduced to strengthen, once again, the role of science communication. Among these approaches, it is important to pursue a more differentiated understanding of target audiences and regional concerns. Science PR has to accept that science itself is no longer the only stakeholder and actor within science communication.

## 1 Introduction

Scholars of communication science have debated the interdependencies between the media and public-relation offices (Altmeppen, 2004; Raupp & Vogelgesang, 2009). Discussion has openly considered whether journalists have turned into Public Relations (PR) professionals or whether press offices at universities and research institutions have already taken issue management (Chase, 1977), agenda-building (Cobb & Elder, 1971), and even journalistic tasks into their own hands (Schnedler, 2011).

Over the last decade science journalism has undergone fundamental changes due to budget and personnel cuts that have led to the closure of many science sections at newspapers (Brumfiel, 2009; Rögener & Wormer, 2017). Particularly here, it can be observed that effective – and despite all the doubts expressed sometimes qualitatively valuable – public relations carried out by universities and research institutions has spearheaded the media coverage of certain areas of science (Rögener & Wormer, 2017; Berg, 2018). Studies on the complex interrelation of PR and journalism (Macnamara, 2014; Williams & Gajevic, 2013; Nelkin, 1995) have shown that PR-dominated science journalism is in fact a reality: "many journalists are in effect retailing science and technology more than investigating it, identifying with their sources more than challenging them"

(Nelkin, 1995, p. 164). The level of influence is inconsistent; Reich, for example, has observed that "studies have attempted to establish a bottom line for PR-originated input, ranging between 25 to 80 percent of news content" (Reich, 2010, p. 799). This might be related to system-specific differences within the national media landscape, as well as the diversity of scientific approaches in the relevant studies (Reich, 2010). Within this context, this study will focus on CCS (Carbon Capture and

Storage) technology.

The analysis of CCS media coverage in German print daily newspapers provides valuable insights and, because the complexity of the subject requires scientific interpretation (Schneider, 2006) in order to enable members of the public to participate in the political discourse. Moreover, CCS is an area that is not only related to technological innovations (IEA, 2009; Oltra et al., 2010) but also to the widely discussed issues of climate change (Kalkuhl et al., 2015) and geoengineering (Anshelm

& Hansson, 2014). Both are attracting more and more attention from scholars of communication science (Buck, 2013; Anshelm & Hansson, 2014; Nisbet, 2009).

Numerous studies have already been conducted on perceptions of CCS (Braun et al., 2017). It seems that in a wide range of countries the public has a largely skeptical view of CCS (e.g., Duan, 2010; Dütschke et al., 2015; Itaoka et al., 2012; Krause et al., 2014). Furthermore, some scholars have observed possible correlation between the framing and acceptance of

CSS (e.g., Duan, 2010; Kraeusel & Möst, 2012; Krause et al., 2014; L'Orange Seigo et al., 2014; Schumann et al., 2014). While it has also been observed that this varies significantly on an international level according to political and social context (Ashworth et al., 2010; Dowd et al., 2014, Pietzner et al., 2011; Terwel & Ter Mors, 2015), there is a lack of detailed analysis of the drivers and actors within media representations of CCS (for an exception to this rule, see Mander et al., 2009).

Nevertheless, the transferability of results from international studies to the German context is limited. While in most

countries CCS is also linked to energy as well as other industry sectors (such as the production of cement and steel; Barker et al., 2009; Wang et al., 2007; Xu & Cang, 2010), the German debate has focused solely on coal (BMBF, 2007; Skrylnikow, 2010). From the beginning, CCS was seen as the savior of the coal-mining industry and energy production through coal (Praetorius & Stechow, 2009). This was perceived by national and international environmental organizations as slowing the much-needed process of winding down coal-energy production (Anderson & Chiavari, 2009). Therefore, the German discourse

about CCS was dominated by strong emotional debates from the very beginning. Research and development programs, such as the German GEOTECHNOLOGIEN program, missed the opportunity to become active and accepted communication partners due to political indecisiveness. Consequently, within the early stages of discourse on the topic, the chance to create an essential factual basis remain unexploited. As things progressed, science only had the chance to react rather than actively integrate itself within the debate.

**1.1 The role of science PR and other actors in CCS-related communication**

Institutionalized public relations of science (science-PR) is driven by the intrinsic intentions of its client (Harlow 1976; Raupp & Vogelgesang, 2009), while at the same time it is singularly focused on transmitting information to its audience. Science PR is often assigned with the role of legitimizing the organisational function of a particular environmental system (Hoffjann,

2007). Science-PR strives to build acceptance by drawing attention to scientific topics and issues (Ten Eyck & Williment, 2003; Schäfer, 2007). As a result, science PR expects science journalism to follow scientific logic and practices, such as scientific ethics and quality management (Weingart, 2003). This leads to the frequently formulated critique from scientists that science journalism has to adapt to meet scientific demands and that internal structural deficits have to be eliminated (Bammé

et al., 1989). On the contrary however, science journalism has to be understood from the perspective of more general journalistic theory (Kohring, 2005). In this light, science journalism has to follow the universal principels of journalism and can be seen as one element of an internally differentiated system that includes other parts such as political, sports, or cultural journalism. Science journalism is thus often more than what science journalists do – and what journalists from other sections do can often be science journalism too (Kohring, 2005, p. 282). As a result, science journalism is characterized by its content.

Science journalism does not serve to get science published or give it prominence, but is rather a service to the general public to enable them to become well informed and participate in democratic decision-making (Luhmann, 1992, p. 633 ff.; Kohring, 2005, p. 282 ff.). Therefore, it is to be expected that journalism which engages with science (or science journalism) takes science into account, even if the dominant focus may well be on other social systems (such as politics, economics, or others). When reading reports about highly scientific and technological issues such as CCS, audiences can expect to be informed not

only about the political or economic features of the topic, but also about the scientific and technological ideas behind it and its principles. Therefore, in journalistic reporting about scientific issues such as CCS, actors from scientific groups should be, if not necessarily dominant, at least present. As such, it seems useful to conduct a study that focuses on the actors in media representations of CCS.

In Germany, four stakeholders can be found within the area of CCS: (1) research institutions (including

universities); (2) energy providers such as Vattenfall, E.ON, RWE, EnBW, and others; (3) political bodies; and (4) non-governmental organizations (NGOs) – such as the Bund für Umwelt und Naturschutz Deutschland (BUND), World Wildlife Fund (WWF) and Greenpeace – and local interest groups (IGs). All stakeholders have individual aims and goals when they become engaged in communicating about CCS, and all take part in the competition for publicity (Malone et al., 2009). Energy providers and political bodies at the national and EU level have tried to promote CCS as a transitional option to

minimize the effects of climate change through the reduction of carbon dioxide emissions (BMBF, 2007; Fischer et al., 2010; Krüger, 2015). Research institutions, while also interested in promoting CCS as a climate mitigation option, have also focus on providing factual scientific knowledge to foster an extensive and open public discourse. Therefore, they seek to attract public attention and foster acceptance by becoming actively engaged in efforts to communicate on CCS (Praetorius & Schumacher, 2009; Chrysostomidis et al., 2013). Both take an active role in CCS-related communication through PR offices

by sending out press releases, conducting public presentations, or pursuing other means to transmit information. In contrast, the political arena in Germany has shown no great interest in contributing content and insight to the debate around CSS. A CCS-dedicated website, conceptualized by the research and development program GEOTECHNOLOGIEN, was ready to be launched but was stopped by political decision-makers (this is based on the author's personal experience as a team member on the website project). Nevertheless, internal struggles within individual parties, as well as disputes between state and

federal policies, have become an important part of the media coverage of CCS (Heisterkamp, 2010). To a great extent, NGOs, which are for the most part not themselves active in CCS research, have demonstrated a predominantly negative attitude toward CCS, for instance preventing governmental investment in CCS research and industry efforts to implement CCS (Schneider, 2017). The allegation that CCS has been misused to improve the image of company can be found in the

recurring argument that the climate-wrecking business activities of the energy providers are being "greenwashed" (Smid, 2009). Taking a closer look at the stakeholder group of NGOs, some (e.g., WWF, 2010) support research into and the development of CCS as a transitional measure that will allow time for better and more efficient measures (Malone et al., 2009). Within this setting, the field of communication science needs to ask whether there are dominant actors in the communication of CCS. The author assumes that (Hypothesis 1) these dominant actors are able to steer the debate in their

favor by setting the framework for CCS, as well as by shaping the public assessment of CCS in line with their own intentions.

Following the observations made by Berinsky and Kinder (2006) that storytelling can guide the audience's own reflections on and interpretations of an issue, modern PR and marketing has highly effectively used storytelling for image building and to increase acceptance of certain issues and products (Sammer, 2014). The PR and marketing work carried out

by companies and NGOs is able to promote individual messages by using complex communication models based on storytelling, as well as issue management and agenda-building techniques. In respect to topics of the utmost importance for society – such as sustainability and climate change – independent science journalism is essential (Nisbet & Fahy, 2015). But while companies and NGOs can use effective PR and marketing strategies, scientific institutions themselves do not usually include communication departments that follow equally high professional standards (Höhn, 2011). While this can be observed

for example at small and middle sized universities, Höhn (2011) also shows, that the level of proficiency in science PR is increasing rapidly. Nevertheless, there are shortcomings, for example in the use of emotion-based framing, in the acceptance of outreach activities by scientists, in communication infrastructure, and other requirements (Höhn, 2011). In addition, we have to take external driving forces such as the ongoing debate about the importance of science communication and science PR or the external framework and context for science communication into account (as done by Murcott & Williams, 2012).

Therefore, the author assumes (Hypothesis 2) that the scientific field, while it can play a significant role in journalistic science communication, is overpowered by professional but instrumentalized PR and marketing by other actors.

## 1.2 Legitimacy and acceptance as driving forces for medialization

Kohring (1997) identifies the need for acceptance as a driving force behind the increased popularization of science. Complementing Kohring's observation, the need for legitimacy can be added as an additional driving factor. Social-science

scholars agree that acceptance is built up by individual risk-benefit assessments (Kraeusel & Möst, 2012; Tokushige et al., 2007; Wallquist et al., 2012; L'Orange Seigo, 2013). Other important factors in increasing levels of acceptance are individually approved opinion leaders, as well as the personal sociopolitical background and life story that guides individual interpretations of communication content (Visscher et al., 2011; Nippa et al., 2014). Consequently, if science PR seeks to increase acceptance

and legitimacy (Jarren & Röttger, 2009, p. 33; Hoffjann, 2007, p. 127), science communication has to be linked to matters of topical relevance to enable communication partners to assess the risk-benefit ratio individually. Science communication has changed in this respect in the past few decades, and medialization can be seen as one result of this change (Kepplinger & Post, 2008; Meyen, 2009). Science communication now has a core focus on highlighting the relevance of science for individuals

and society (Herrmann-Giovanelli, 2013, p. 65f). Nevertheless, this is not sufficient to further increase acceptance and legitimacy. Affective attitudinal components cannot be fully controlled by science communication, but they are of the utmost importance for building up acceptance and legitimacy (Finucane et al., 2000). The following analysis of media representations of CCS therefore seeks to locate observable aspects that will help to identify mechanisms of acceptance and legitimacy building. The importance of individual risk-benefit assessments is one such observable aspect; a strong journalistic focus on

risk-benefit ratios, the emphases of opinion leaders, and the clear integration of emotional language serve as indicators of an intended acceptance and legitimization approach.

## 1.3 The organization of CCS-related communication in Germany

The scientific field – and within the thematic framework of CCS this essentially refers to the earth sciences – is partly populated by communicators and communication tools that possess limited professional standards (Höhn, 2011). Still, only a

relatively small number of research institutions maintain professional outreach offices specialized in earth sciences to communicate complex and multilayered topics. At the same time, energy companies, while they contain well-staffed professional communication offices, have exhibited only low-level enthusiasm for outreach related to CCS. Their reticence has been motivated by the fickle actions of and lack of support from the third stakeholder group: the political arena. Without a fixed legal framework for investing in CCS, companies have understandably shied away from engaging in public debates

about it. There has been a lack of CCS-related communication triggered by politics because of internal disagreements within parties and between the state and federal political levels (Heisterkamp, 2010). The effectiveness of CCS-related communication has been demonstrated by NGOs in relation to projects that were planned but later abandoned in Hürth (North Rhine-Westphalia) and along the Schleswig-Holstein coast, as well as the wave of protests that accompanied CCS projects in eastern Germany. Here, NGOs such as Greenpeace and the BUND used established tools to engage the public in

their strong campaigns against CCS. One of the most successful models for achieving this included the use of powerful frameworks focused on emotions in CCS-related communication. The 2013 article by Greenpeace titled "Death from the Chimney: How Coal-Powered Energy Ruins our Health" ("Tod aus dem Schlot – Wie Kohlekraftwerke unsere Gesundheit ruinieren"; Greenpeace, 2013) provides an example of the utilization of the emotional framework.

Science communication has meanwhile faced a dilemma: as part of the scientific tradition, science communication

that originates from science itself (science PR) is strongly aligned with factual information rather than emotion. Even more problematically, Dunwoody and Peters address a potential "systematic misconception of the recipients' interests" (Dunwoody & Peters, 1993, p. 334) by the scientific field. Appeals to emotion as well as storytelling techniques are used by

the media (science journalism) while the cognitive components of communication are neglected in favor of affective communication. Since recipients can decide individually whence to get their information, journalistic representations of science have become the favored source to obtain factual information, since affective communication deals with topicality. If the scientific field were to switch to using emotion-based communication, familiar communication patterns would be

abandoned. How this would affect levels of acceptance and the legitimacy of science among the public cannot be foreseen; therefore, science is trapped in "emotionless communication" behavior. At the same time, the media follows internal systems of logic that are resampled in the news value model (Galtung & Runge 1965; Kepplinger & Ehmig 2006). The selection of "newsworthy" content obviously results in an overemphasis on risk that overrides the factual communication of science. The transformation of cognitive information into affective communication is boosted by science PR. Because science PR is in

competition with PR efforts from other societal arenas, such as politics, sports, the economy, and others, it seems reasonable for science PR to use selection processes similar to those of the media. This increases science communication's focus on risk and benefit, and on demands and expectations, while the recipients expect factual and research-based information from science (e.g. Maier et al, 2016) to inform their individual interpretative and decision-making processes.

## 2 Analysis design

To gain a better understanding of the role of science PR within the media coverage of CCS in Germany, a long-term case study was conducted, covering daily newspaper articles from January 2004 to December 2014. This time frame begins with the substantial funding of CCS research and development projects by the German Federal Ministry for Education and Research (BMBF) and ends with the month that followed the final decision upon CCS law in Germany.

      The data used for the analysis was taken from a media database that contains about 120 million articles from German

daily print newspapers.[1]

      To get a representative sample, the online accessible archive was searched for the keywords "Kohle" (coal) and "CCS."[2] The keyword CCS was selected because of its widely established use in the scientific and political arenas. The keyword *Kohle* was used due to the introduction of the German CCS debate through a prominent statement by the NGO Germanwatch in 2004 (Dukat et al., 2004), which directly related CCS to the coal-mining industry. At a later point, CSS in

Germany was viewed in close relation to the production of energy through the burning of coal, and the German federal government also framed its CCS strategy around the coal-mining industry (Heisterkamp, 2010). Due to the long history of the coal-mining industry both in West and East Germany, other industry-based links, such as those between CCS and the production of steel and cement have not featured in the German public debate around CCS.

---

[1] The wiso Press database at the Freie Universität Berlin in Germany was used. This database included more than 120 million articles by German daily newspapers within the time frame of our analysis. More information about the database and the sources and titles that are included in it can be found on the GENIOS website (www.genios.de).

[2] Because of the frequent use of CCS as an abbreviation for the Congress Centrum Suhl (Suhl Convention Center), a method to exclude Suhl was formulated within the search term: (CCS AND Kohle*) NOT Suhl.

The utilization of this simple, first selection process resulted in a list of N = 5,150 articles. One hundred and ninety-two articles were deleted from the sample due to their international origins. This is based on the fact that the author lacked in-depth insight into the political and scientific environments of other countries, and regional influences on the relevant media coverage could not be determined in detail. Eighteen press agency articles were also deleted from the list, since the analysis is designed to focus on stakeholders and their influence on the coverage. Taking press agency releases into account would bias this analysis in favor of said agencies. Because of the setup of the database used, various examples of double posting were identified and deleted. The author also deleted articles of less than one hundred words in length (mostly event notes), letters to the editor, and commentaries. The resulting list of $n_{basic}$ = 2,809 articles is called the basic list and was used for headline analysis.

To conduct a qualitative content analysis, a further reduction through the application of a temporal filter was necessary. This temporal filter consisted of a quasi-week sum of articles for each day. This quasi-week sum resulted from adding the number of articles from one day to the number of articles for both the three previous and the three subsequent days. Consequently, the reduction was based on the concept of reducing the number of artifacts and biases due to individual events or dossiers (an overview of a topic from one newspaper that consists of many articles with different foci). Weekly artifacts, such as science-related issues for a single day or weekend, were also reduced through the quasi-week sum approach. This is necessary to reduce effects by thematic dossier series (a set of up to 10 articles about a single issue, mostly written by the same author or team of authors, published in a single outlet). These dossiers, which can be seen as singular (one day) publication maxima, would bias the analysis of the regional distribution of articles as well as of the temporal evolution of the topic. All articles from days with a quasi-week sum of equal or more than forty were included within the quantitative content analysis. A cut-off at forty articles was chosen to include the ten peak periods, which reduced the number of samples to a manageable but representative size that still covers the different phases of the issue-attention cycle (Downs, 1972).

[Figure 1]

*Figure 1: Quasi-week sum plotted per day. All articles for days with a quasi-week sum of forty or more (solid line), as well as all articles from May 2007 and February 2013 (solid circles), were included in the analysis.*

In addition, the period of greatest publication on the topic from 2007 and 2013 was included within the analysis in order to also get publications from the first and last phases of the issue-attention cycles. After applying this temporal filter, $n_{filter1}$ = 569 articles (about 20% of the basic list) were analyzed in the qualitative content analysis.

[Figure 2]

*Figure 2: Sampling process: In the first step, all articles collected in the media database (wiso) were scanned for keywords; in the second step, these articles were evaluated for double postings, media agency releases, and international publications; in the third step, the resulting list was filtered using a temporal filter; finally, a randomized sample was taken for a detailed actors' analysis.*

# 3 Analysis and results

According to Schäfer (2008) and Marcinkowski (2015), medialization of a topic is recognizable via three indicators: extent, plurality, and a high level of controversy. With more than 5,000 articles in ten years, media coverage of CCS can be described as extensive. Taking weekends into account, CCS has been the subject of newspaper coverage to the extent of an average of
1.6 articles a day (3.5 articles in the month of the most extensive media coverage). Furthermore, the regional extent of coverage can be shown by looking at the newspaper titles and their regional distribution. Eighty-nine titles (individual newspapers) covered CCS; these were distributed throughout Germany. About 19% of articles about CCS were published in nationwide publications. As a result, the indicator of extent can be observed both on a temporal and on a spatial plane.

Since plurality as well as the level of controversy can only be determined through a quantitative content analysis, the
following sections are dedicated to these indicators.

## 3.1 The thematic plurality of CCS media coverage

The temporal evolution of CCS coverage was predominantly driven by political developments within this ten-year time frame. Without the recurring political debates about a CCS law in the German Bundestag, CCS would not have been given such extensive media coverage. Nevertheless, the quantitative analysis demonstrated that non-political perspectives related to CCS
were able to set the media agenda – at least for a few weeks – as well.

The indicator of plurality, which shows a certain extent of medialization of the issue, can be observed by looking at the thematic evolution of the topic in German newspapers in detail. In early reporting on CCS, journalists focused on events such as the Durban Climate Change Conference in 2011 (COP17). These events provided an entry point for CCS to gain coverage in the media, which are accompanied by the few overview articles that can be found in the ten-year time frame of the
study. Most of these articles are also closely related to climate protection frameworks that quickly disappear to make way for those related to technological development and pioneering ideas attributed to the participating German industries. The opening of the pilot plant in Spremberg (Schwarze Pumpe, Brandenburg; September 9, 2008) can be seen as the endpoint of this period, which produced only a few articles on the CCS technology itself. Only six months later, CCS media coverage began to focus on political frameworks. The debate and controversies that surrounded the first (June 2009) and, later, the second draft of a
German CCS law (September 2011) dictated the journalists' approach toward CCS for nearly two years. In 2011 the Federal Institute for Geosciences and Natural Resources (BGR) decided not to publish a study about geological sites with storage potentials for carbon dioxide in Germany (February 2011). This was used by NGOs (dominated by Greenpeace) as an opportunity to publish media releases that focused on the societal responsibility of CCS, and the media coverage became attentive to energy providers who were active in commercial CCS research and development (Vattenfall, RWE, E.On, EnBW,
and others). Surprisingly, the issue of withholding a study about potential storage sites did not generate as much media attention as one might have expected. The reasons behind the decision not to publish the study were not explored by journalists. Furthermore, existing critique of the BGR, which focused on cooperation with industry partners such as RWE and others

(Goerne et al., 2010), was not part of the media coverage in print newspapers (but was highlighted in numerous blogs and websites; LobbyControl, 2011). This shows that this particular scientific study was not considered to be particularly important by journalists throughout Germany. Subsequent publications used – once again – the political framework of controversies and debates to cover CCS. This changed at the end of 2011 with Vattenfall's announcement that it had stopped CCS-related research and development in Germany. After this announcement, economic and social frameworks became dominant in newspaper articles, mixed with those centered around energy politics, labor-market policy, and the projected end of Germany's use of coal as an energy resource. In mid-2012 media coverage switched back to political frameworks related to the final decision on a German CCS law (June 2012). For the first time regional demands and expectations became dominant. As some states in Germany had already asserted that they would not allow CCS within their borders, others quickly followed suit. This led to the de facto death of CCS utilization in Germany.

This outline of the thematic evolution of CCS coverage demonstrates pluralization. Moreover, the use of a political framework that is aligned with conflicts within parties and between state and federal policies shows that there are high levels of struggle over a topic that originates from the scientific field. Therefore, the third indicator for medialization can be observed as well. As the media coverage of CCS is dominated by political frameworks, the observation of medialization cannot be attributed to science. Because science obviously does not play a major role in the media coverage of CCS, but politics does, it seems justified to suggest that the analysis of the coverage of CCS reveals the medialization of politics. Given that public acceptance and legitimization are even more relevant for politics , this is not surprising. Due to its technological and geo-chemical complexity as well as in respect to the focus on research driven feasibility studies such as Schwarze Pumpe, Ketzin and others, a more intense reference to relevant science was to be expected.

What is surprising is that a scientific topic can be dominated by the medialization of politics and that scientific actors have to step back in favor of political actors. To identify whether this transfer of actors has implications for the coverage of a scientific topic, the following section will take a closer look at those actors and how they shape the coverage of CCS in German newspapers.

[Table 1]

*Table 1: Important dates for CCS research and development, and CCS policy in Germany. The number of newspaper articles dealing with these events is given.*

It is noticeable that, despite its significant relevance for public discourses about sustainability and climate change CCS nearly completely vanished from the media agenda only days after the final decision about a CCS law was reached. While research on CCS and in the pilot plant Schwarze Pumpe, where CCS was tested on an industrial scale, was carried out for a number of years after 2012, journalists no longer saw any reason for media coverage. This first impression allows us to conclude that science on its own did not have the means to influence the media agenda. This is also supported by a descriptive analysis of the key thematic elements represented in the media coverage.

Six key aspects were responsible for more than 80% of the media coverage. Because of the unprecedented indecisiveness of the German government (see also Figure 1), the key thematic aspect of *CCS law* dominated the media coverage in German daily newspapers as expected. The sudden exit of Vattenfall from commercial CCS research and development (*CCS Exit*), overview articles on the relevance of CCS to power production and the coal industry as well as science and research (*Overview of CCS*), the political stance and discourse on the level of the federal states (*Political Stance*), the public and political debate about energy politics (*Energy Politics*), and, finally, media coverage of the launch of the Schwarze Pumpe CCS pilot power plant (*Pilot Plant*) comprise about one third of articles about CCS. All other themes make up one fifth (20%) of all articles.

[Figure 3]

*Figure 3: Distribution of the key thematic aspects (569 articles).*

Four of these aspects (*CCS Law*, *CCS Exit*, *Political Stance*, and *Pilot Plant*) correlate directly with the temporal evolution of CCS. Another key aspect is *Energy policy*. None of the key thematic aspects are directly related to science or technology. This shows that journalists did not hold the scientific field to be the only relevant societal system within the area of CCS. Instead, the focus on energy policy shows that journalists view the economic and the political arena to be the most relevant stakeholders.

While 80% of the media coverage is dominated by six key aspects, the remaining 20% is divided among 20 other aspects. These cover areas such as CCS in relation to fracking or the anticipated displacement of small villages due to increased coal-mining activities. Taking this broad portfolio of key thematic aspects into account, plurality, at least to some extent, can be identified within the media coverage of CCS.

## 3.2 Actors and processes of agenda-building

All individual persons and institutions were counted as actors, as long as they were mentioned in the body text of the articles (this excludes headlines). Since intense research about the positions and functions of the actors was necessary, this detailed analysis was conducted through a randomized sample of $n_{random}$ = 255 articles (10% of the basic list). The sample was tested to resample the distribution of publication titles found in the basic list.

Two hundred and forty-nine individual actors were identified, who were mentioned 1,050 times altogether. The energy provider Vattenfall dominates this list with 249 mentions. When the 67 mentions of Tuomo Hatakka, who was the country chairman of Vattenfall in Germany at this time, are also added to this list, it accounts for roughly 20% of all actor mentions. This is surprising because Vattenfall is not the only energy provider who has been involved in CCS in Germany. Others, such as E.ON, RWE, and EnBW, began work on CCS projects as well; they also gained a lot of attention from NGOs and were targeted by protests from IGs. Nevertheless, all three account for only 55 mentions (5%) within the overall total. By looking at the articles that mention energy providers more closely, one can see that about 47% of them communicate in a positive or highly positive manner about CCS. Only 21% of these articles demonstrate a negative or highly negative attitude

toward CCS. Here positive and negative are defined through the use of a keyword with positive or negative connotations as well as based on the analysis of the used frames. The framing analysis followed a concept by Goffman (1974) and is based on the idea, that the communication about a topic can be shaped according the intended meaning. Frames therefore are meaningful contexts for communication, that guide the perception and interpretation attributed to words or phrases (the current debate about framing is summarized by Cacciatore et al, 2015). To define positive and negative associations for keywords, this study follows a concept elaborated by Ungerer (1997). He observed that emotions can be divided into those that are directly mentioned and indirectly caused emotions. While the latter are only observable through audience evaluations, the former can be operationalized in content analysis. The direct use of phrases such as "War on the climate," the indirect creation of an image through the use of biased frames, and the use of metaphors and allegories provides evidence for forms of manipulation (whether intentional or unintentional). This content analysis studied the direct use of strong emotional keywords (positive, e.g.: opportunities for climate conservation, career opportunities; negative, e.g.: climate killer, toxic technology, world in turmoil) and an overall frame analysis (positive = CCS as a tool to help mankind solve the challenges posed by climate change; negative = the research and development of CCS is only driven by commercial interests). The observation of how and to what extent positive and negative denoted frames are used is important: scholars of communication research have shown, that negative framing has stronger influences on the perception of a theme that positive framing (e.g. Kahneman & Tversky, 1979, Lau, 1985, Cacioppo et al 1997, de Vrees er al. 2011) It should also be noted here that the concept of emotion and the presumed emotional content and connotation of words and terms, are strongly dependent on the particular sociocultural context (Wierzbicka,1995). Therefore, the operationalization and interpretation of emotions in newspaper articles are influenced by the sociocultural background of the person(s) assigned to do the coding. As the coding in this study was undertaken by one person, systematic bias can be neglected. Still, the interpretation of words and terms in respect of their emotional content is somewhat vague. Therefore, only strongly emotional words and terms (for example, "war" and "killer" as negative or "chance" and "opportunity" as positive) in regard to their textual contexts were coded and interpreted.

Matthias Platzeck, Minister President of Brandenburg from 2002 to 2013 (56 mentions), and Ralf Christoffers, Minister of Economic Affairs and European Affairs of Brandenburg from 2009 to 2014 (36 mentions), account for 8% of all actors mentioned in CCS articles. While both were advocates of CCS in Brandenburg, both were also connected to negative communication about CCS. Forty-seven percent of articles that mention Platzeck and eighty-seven percent of articles that mention Christoffers demonstrate negative to highly negative attitudes toward CCS. This observation is based on internal conflict within the Social Democratic Party of Germany (SPD). In contrast to the SPD at national level, the SPD in Brandenburg at state level approved of CCS and coal mining.

The only actor from the German scientific community cited within the media coverage of CCS was the BGR with 31 mentions (3% overall). The only individual scientist named in articles about CCS was Ottmar Edenhofer, an internationally renowned expert on climate economy at the Potsdam Institute for Climate Impact Research (PIK). He was mentioned six times. Other science-related terms such as "scientist," "science team," "researcher," "climate scientist," "geologist," or even "expert"

accpunt for 114 mentions (11%) altogether. The content analysis shows that 66% of all the articles that mention actors from the scientific field are connected to positive communications about CCS.

[Table 2]

*Table 2: Distribution of connotation trends in relation to the main stakeholder groups (255 articles).*

While the detailed analysis of actors in the media coverage of CCS reveals the striking dominance of economic and political actors, this result cannot be unquestioningly transferred to the whole of CCS newspaper coverage. Since the media coverage of CCS correlates strongly with the political evolution of the topic, and because CCS is, as a climate-change mitigation option, not only a scientific but also a societal, economic, and political issue, the obvious dominance of actors from the political and economic arenas has to be expected. Nevertheless, integrating the observation of this correlation with results

from the analysis of actors shows that scientific events and input have no impact on the media coverage. Neither the storage site assessment conducted by the BGR nor well-regarded international conferences, such as the CCS Status Meetings conducted by the research and development program GEOTECHNOLOGIEN, which included special presentations and press-conferences, were able to attract journalists' attention toward CCS.

      Of great importance here is that the analysis of actors in the newspaper coverage of CCS in Germany shows striking

differences in the attitudes toward CCS that were communicated. While articles featuring actors from politics took a slightly negative stance toward CCS, articles focusing on NGOs and IGs were strongly dominated by the negative communications of the topic. In contrast, articles featuring actors from an economics background are dominated by positive attitudes toward CCS. The analysis also supports the conclusion that articles featuring actors from the scientific field are also dominated by the positive communication of CCS.

[Figure 4]

*Figure 4: The distribution of actors by stakeholder group as represented in German daily newspaper articles about CCS.*

      The results of this mixed quantitative-qualitative analysis are surprising because it would be expected that by enabling the public to participate in an open and transparent discourse on a topic such as CCS, which has great social and ecological relevance, the resulting discourse would be dominated by the field of science and scientific approaches. Nevertheless, the

political arena shapes the journalistic coverage of CCS in German newspapers. It seems that conflicts within the political realm are considered by journalists to be the most newsworthy. Therefore, in support of results from an earlier study by Pietzner et al. (2014), the dominant focus in German newspapers is on politicians and disputes between them over CCS. Scientific perspectives on CCS are considered to be even less important than the positions of NGOs and IGs. Furthermore, despite observations that science PR is on the rise (e.g. Meyer, 2010, Murcott & Williams, 2012, Trench, 2017), it seems that the

scientific field is unable to establish itself as a relevant source of information for journalists. Whether this is because of limited resources for outreach within scientific bodies or due to a misconception of how journalists seek out their sources remains unclear. Nevertheless, these results support the findings of Trumbo, who analyzed media coverage of climate change in the US:

The more alarming aspect found within the results of this study is that, relatively speaking, scientists left the debate as it heated up. In fact, scientists found themselves sharing a shrinking proportion of growing media attention during an important stage of the public debate over climate change (Trumbo, 1996, p. 281).

### 3.3 The lack of a reason for the weaknesses of science PR

As the basis for identifying reasons for the weakness of science PR in relation to other social systems, this study took the concept of functionally differentiated societies into account. Holzer (2011) observed, strongly influenced by Marcinkowski (1993) and others, that there is no hierarchy within societal systems (Hoffjann, 2007). While self-observation is conducted among the public through descriptive medias (Hoffjann & Arlt, 2015; Marcinkowski, 1993), critical journalism is also used as a service that allows a second level of observation that finally leads to efficient and constructive self-imaging. This process

becomes even more complex since functional systems try to actively interact with journalism through the use of system-immanent PR efforts. Thus, all functional systems in fact sustain their own PR bodies – even journalism supports media PR that finally leads to journalism about the media itself (Hoffjann & Arlt, 2015). Focusing on the relation between journalism and science again, the previous discussion suggests that science journalism – the part of the journalistic field that deals mostly with science – does not serve science but rather society (Luhmann, 1992.; Kohring, 2005). Consequently, journalism does not

seek to respond to the demands and expectations of science but to those of society. Therefore, the scientific field needs to realize that journalism does not work according to the demands and expectations of science. What is more, science journalism, in responding to the demands of society, does not focus on science at all but on scientific topics relevant to the public agenda. It is not organized around what science wants journalism to communicate. Instead, science journalism is founded on the external observation of science. In other words, science journalism is not all about science.

Science PR has to recognize that science journalism will not communicate issues and themes that originate from the scientific field alone. Science journalism will always seek to focus on the relations between science and society and other functional systems. As long as science PR is trapped in the conception that science journalism serves science as a means of communication rather than seeing science journalism as a way to emphasize scientific social relevance, science PR will fail to attract attention on a large scale. There needs to be a shift within science and science PR to help change the image of journalism

into one of a relationship manager rather than a service provider. Within the CCS context, this shift has not occurred. Instead, other areas of society, such as politics and economics, have filled the gap between science and society and thus dominate the journalistic coverage of CCS.

### 4 Conclusion

In contrast to previous studies about medialization in the field of climate change (Nisbet, 2009; Schäfer & Schlichting, 2014),

neither internal scientific conflict nor scientific uncertaintyplayed a significant role in newspaper coverage of CCS. In fact,

and verifying Hypothesis 1, this study shows that dominant actors who shaped the media representation of CCS and that media coverage of CCS was uncoupled from science. It was rather the case that, verifying Hypothesis 2, the coverage was strongly linked to politics and economics. As journalists, following their self-imposed ethical guidelines to reveal sources of information (Deutscher Presserat, 2008), use their sources as evidence of their serious and responsible journalistic work, the lack of
scientists as named sources can be interpreted as a failed conversation between journalists and scientists. This might be related to a misconception, namely science's failure to recognize that the services and functions of journalism are directed toward society rather than science itself. In addition, it seems that the lack of emotion in science PR might be another obstacle. While science itself demands that science PR follows scientific logic (for example, focusing on verifiable factual information), NGOs and IGs prefer emotional communication.

Without a paradigm change within science PR, journalistic communication will focus on actors from NGOs and IGs because of the greater newsworthiness attributed to emotions and conflict. Since a focus on NGOs, IGs, and conflicts within politics implies highlighting negative attitudes toward CCS, media coverage of CCS will also become dominated by negative positions on the topic. In fact, 41% of all the studied articles published about CCS in German newspapers show this effect. This observation contrasts with studies that show how innovative technology and research is predominantly communicated in
positive ways (Weaver et al., 2009).

If science is to reassert its role as an established, reliable, first-choice partner for science journalism, it has to increase its own understanding of the need to forge strong and diverse links with other functional systems within society. Science PR has to open up to the use of emotions and the highlighting of social relevance (for example, by focusing on the scientists rather than the science itself). Otherwise, different systems within society will increasingly dominate the presentation of science,
leading to a disproportionate emphasis on an external and heteronomous image of science. Narrative frames have to be established that allow society to perceive relevance in research and development. Nevertheless, the strategic utilization of narrative frames has to be carefully devised:

When frames are conceived as given, the role of communication is seriously constrained as they can only convey their
message within the cultural framework of the target audience. When frames are conceived as dynamic, communicators can intervene in the contest of frames either by modifying a communication frame or creating a new interpretation of reality. (Olmastroni, 2014, p. 12f)

Frames used by science do not have to target emotions alone: they have to respond to the demands and expectations
of the recipients as well. If science can foster links with the individual environments of the recipients on a sociocultural level, it will be able to demonstrate its everyday relevance even more successfully. This analysis of media coverage of CCS in German newspapers has shown that media coverage is already at the point where science has been replaced by other systems within society as the primary communicator of scientific topics. Science – and science PR – has to accept that it does not have the unique right to talk about science.

**Acknowledgement**

This publication has benefited from valuable input by Prof. Alexander Görke from Freie University Berlin and Prof. Armin Scholl from the University Munster. I would also like to take this opportunity to acknowledge the time and effort devoted by the reviewers to improving the quality of this article. I also acknowledge the support of the Open Access Publishing Fund of University of Potsdam.

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

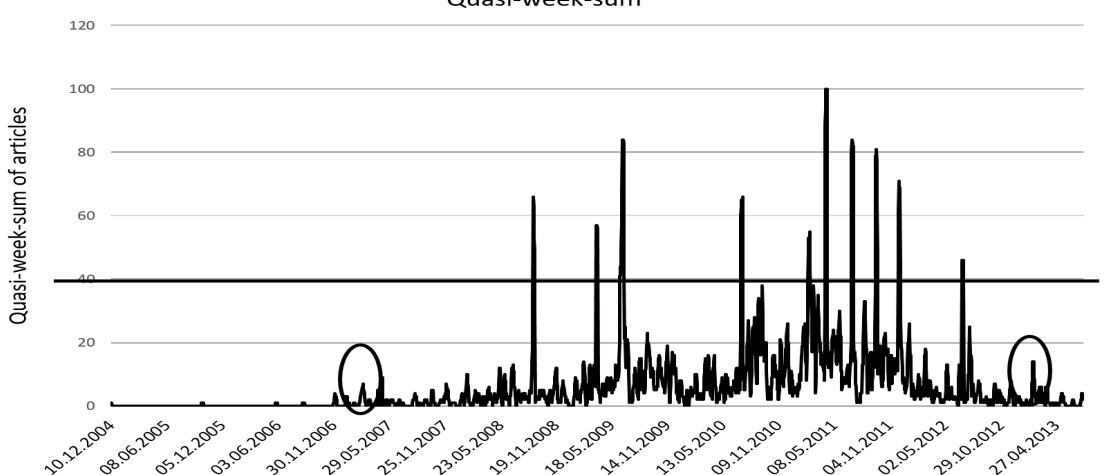

*Figure 1: Quasi-week sum plotted per day. All articles for days with a quasi-week sum of forty or more (solid line), as well as all articles*

5  *from May 2007 and February 2013 (solid circles), were included in the analysis.*

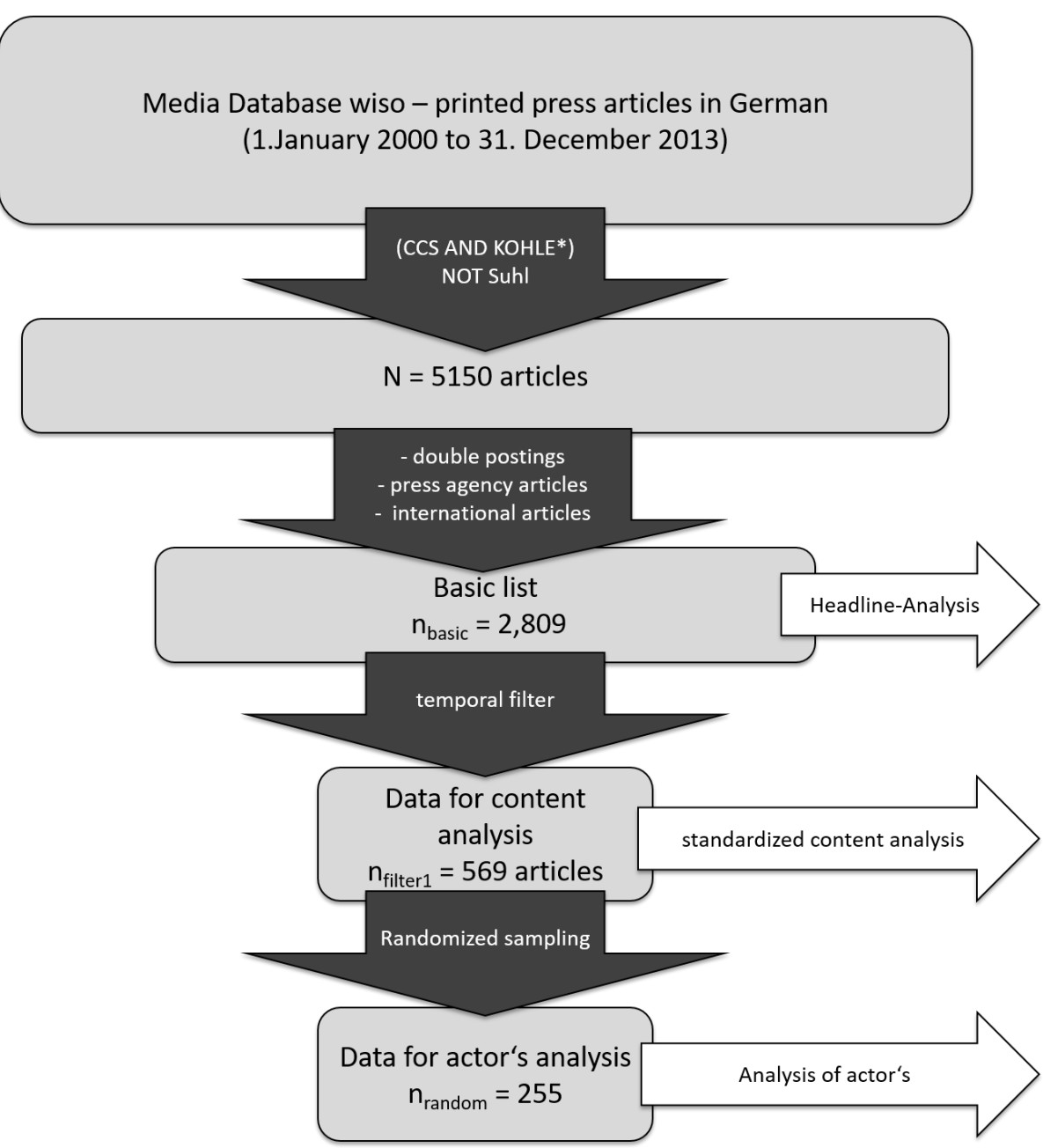

Figure 2: Sampling process: In the first step, all articles collected in the media database (wiso) were scanned for keywords; in the second step, these articles were evaluated for double postings, media agency releases, and international publications;

*in the third step, the resulting list was filtered using a temporal filter; finally, a randomized sample was taken for a detailed*

| | |
|---|---|
| **May 5 and 6, 2007** | • Publication of the Fourth Assessment Report (AR4) by the IPCC<br>6 Artikel |
| **September 5 to 12, 2008** | • Start of the CCS pilot power plant Schwarze Pumpe<br>54 Artikel |
| **March 30 to April 5, 2009** | • German federal cabinet decides on the first draft for a national CCS law<br>44 Artikel |
| **June 15 to 29, 2009** | • German federal parliament (Bundestag) cancels further debates on the first draft of the CCS law<br>16 Artikel |
| **July 12 to 17, 2010** | • Brüderle/Röttgen (German Energy and German Environmenal ministers) submit a second draft of a German CCS law<br>58 Artikel |
| **February 14 to 21, 2011** | • BGR study about geological sites with storage potential for carbon dioxide in Germany becomes public<br>51 Artikel |
| **April 11 to 16, 2011** | • German federal cabinet decides positively about the second draft for a national CCS law<br>84 Artikel |
| **July 5 to 11, 2011** | • German federal parliament (Bundestag) decides positively about the second draft for a national CCS law<br>67 Artikel |
| **September 21 to 27, 2011** | • Federal Council (Bundesrat) declines second draft<br>70 Artikel |
| **December 2 to 15, 2011** | • Exit from commercial CCS research and development by Vattenfall<br>75 Artikel |
| **June 27 to 30, 2012** | • Mediation committee (Vermittlungsausschuss) recommends changes to the CCS-law, Bundestag und Bundesrat accept changes<br>35 Artikel |
| **February 11 to 13, 2013** | • Federal Government of Germany is asked for an official statement on fracking<br>9 Artikel |

*actors' analysis.*

*Table 1: Important dates for CCS research and development, and CCS policy in Germany. The number of newspaper articles dealing with these events is given.*

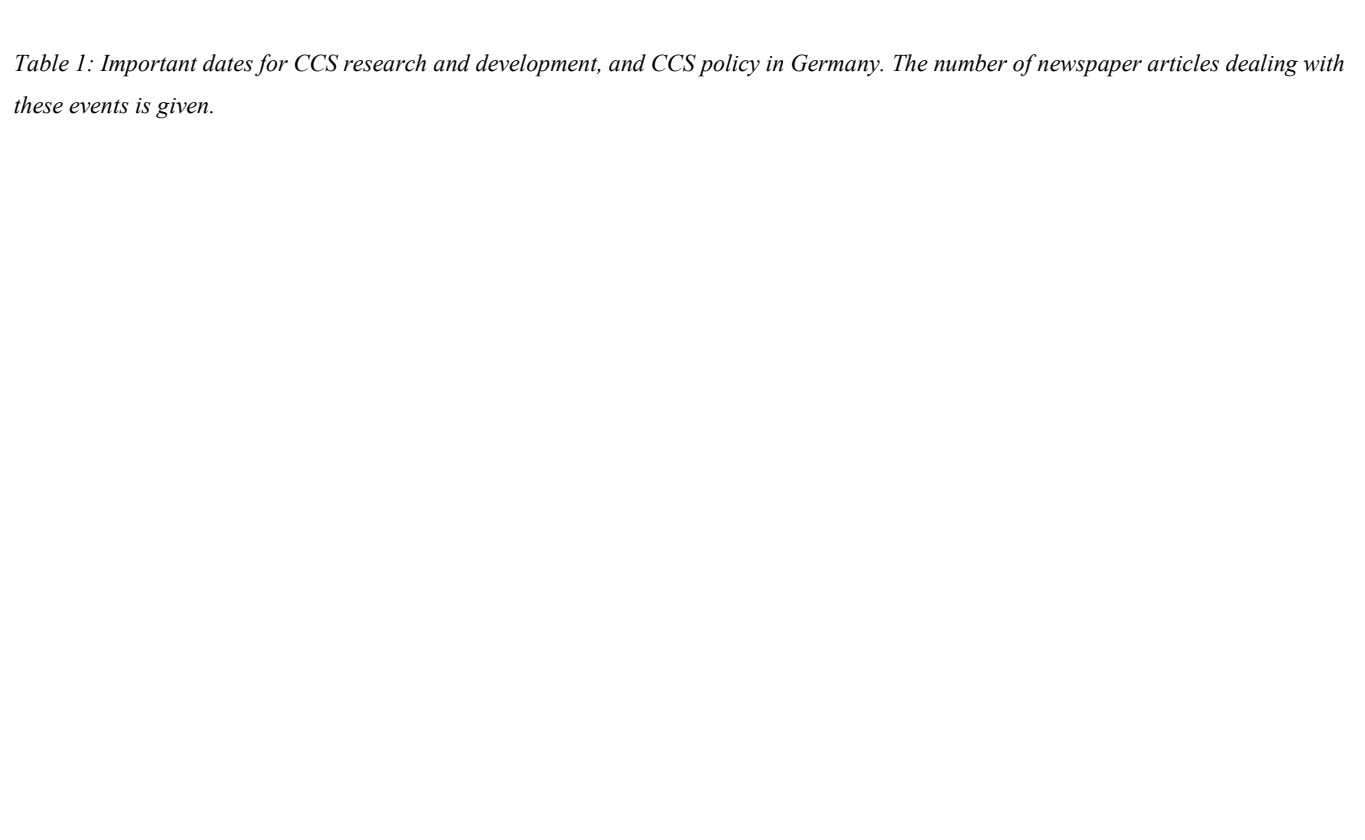

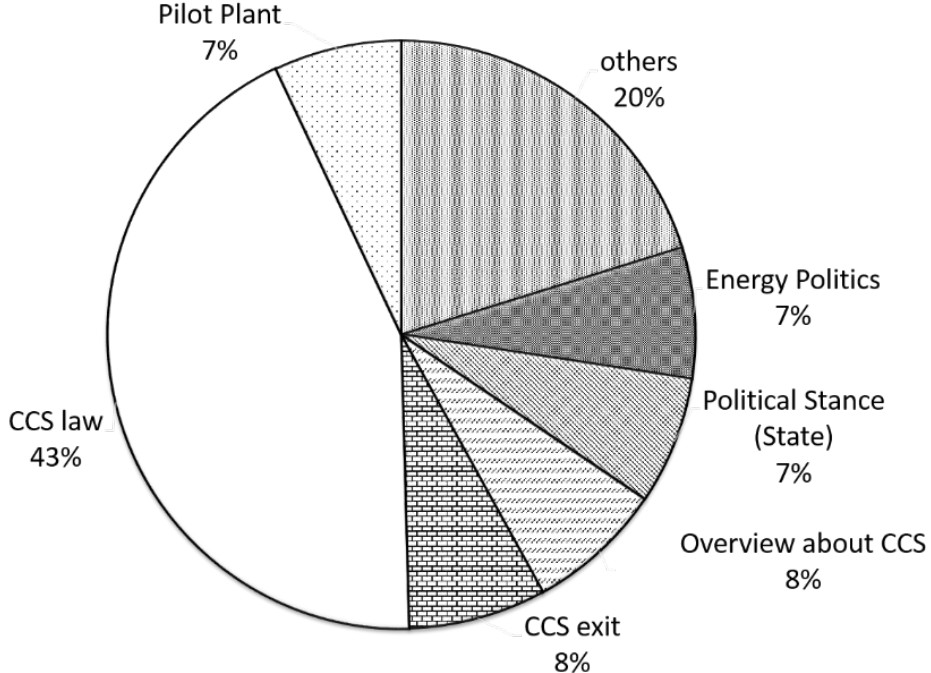

5    *Figure 3: Distribution of the key thematic aspects (569 articles).*

| | highly negative | negative | neutral | positive | highly positive |
|---|---|---|---|---|---|
| **Politicians** | 16,0 % | 31,3 % | 22,9 % | 23,6 % | 6,5 % |
| **Scientists** | 6,7 % | 20,0 % | 6,7 % | 46,7 % | 20,0 % |
| **NGO representatives** | 35,5 % | 38,7 % | 12,9 % | 12,9 % | 0,0 % |
| **Industry representatives** | 1,3 % | 14,3 % | 35,1 % | 45,5 % | 3,9 % |
| **overall** | 13,5 % | 26,6 % | 24,3 % | 29,9 % | 5,6 % |

*Table 2: Distribution of connotation trends in relation to the main stakeholder groups (255 articles).*

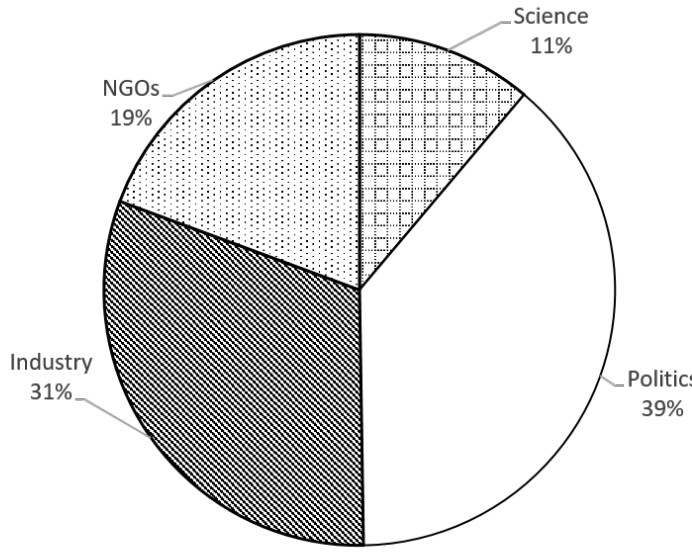

*Figure 4: The distribution of actors by stakeholder group as represented in German daily newspaper articles about CCS.*