# Peer review of "The Takeover of Science Communication: How Science Lost Its Leading Role in the Public Discourse on Carbon Capture and Storage Research in Daily Newspapers in Germany"

_Geoscience Communication, 2018_

## Referee Comment (RC1) · J. Roberts (Referee) · 10 Jul 2018

Re: Research article "The Takeover of Science Communication – Science Lost its Leading Role in the Public Discourse of Carbon Capture and Storage Research in Daily Newspapers in Germany" by Simon Schneider.

GENERAL COMMENTS In this research article the author presents a content analysis of newspaper headlines about Carbon Capture and Storage and Coal published in Germany between 2004 and 2014. A filtered cohort of articles are selected for thematic

analysis. The author finds that media coverage of CCS is decoupled from science, and is more strongly linked to politics and economics. The author concludes that science PR needs to step up its game if it is to be represented in science journalism. The approach, argument and outcomes of the paper could be of interest to an international and interdisciplinary audience, but some significant improvements must be made to make it accessible to these audiences. For example, the data analysis and presentation needs to be much deeper and much more transparent, and the manuscript content needs to be presented more clearly for non-German audiences and to a higher quality. Further, the title is misleading; there is nothing in the manuscript to suggest that science ever had a leading role in CCS discourse. Finally, there is a fundamental flaw in the assumption that CCS + Coal is, or should be, a predominantly scientific topic, and not a political or economic, or environmental and social justice issue. For this reason, I recommend publication, but subject to major revisions to the article and its framing.

SPECIFIC COMMENTS 1. The analysis is specifically for COAL + CCS. This isn't mentioned in the abstract, or introduction, until the analysis design is outlined. Why are you selecting coal? Presumably because coal + CCS is the predominant conversation in Germany, rather than gas + CCS, or cement/steel works + CCS like the discussions in the UK and elsewhere. But the rationale needs to be stated explicitly, and CCS in Germany should be contextualised in the introduction much more clearly. It may also be worth considering how filtering for coal headlines might this affect the results. But here lies one of the major flaws in this paper. Coal extraction and combustion is laced with issues around pollution and human health, climate change, continued industrialisation of the landscape and the ethical and moral discussions that arise from this – including the displacement of communities, not to mention employment, economic wealth and so forth. In addition, CCS is not necessarily a topic for science. Sure, there are scientific fundamentals behind CCS, as the author correctly states on page 3 line 1. But, the not only is the science of CCS not particularly novel or innovating (the science/engineering components of CCS have individually been around for a long time, and so are not brand new exciting science), CCS is a climate change mitigation

Interactive
comment
technique. While climate change is, at its core, a scientific issue, the solutions are not scientific; they are societal and economic and political etc. So, why would we expect CCS to be predominantly the domain of science journalism? The author needs to re-consider this assumption, and the framing of the paper to reflect this. 2. The author doesn't satisfyingly explore in the article (ie in the introduction, discussion or conclu-sions) what the purpose of science PR is, and whether the purpose or role or intentions are affecting how it is then portrayed in the media. If the purpose of Science PR is to "inform" as hinted at on page 5, then is it problematic that CCS journalism doesn't hugely cover science/scientists? What is the author wanting to see instead? A report on CCS policy and the progress in CCS law with a commentary from a scientist about CCS? This is a slightly complex and thorny issue which isn't currently very well consid-ered in the manuscript. 3. It is not clear what media the publications analysed were, are these online and/or print articles, are they hosted/written by news organisations, topic specific organisations (e.g. Carbon Brief) or e.g. by interest groups? Are they opinion pieces? How might these different article types influence the results? Further, why was a cut off of 40 articles chosen? Please give the rationale, even if it is simply arbitrary. And why isn't the horizontal line in Figure 1 at 40, not at ∼35? It is also not very clear about whether the headline was analysed, or the article content, and how (i.e. what tools were used). I am assuming the content was analysed because the author then goes on to talk about actors specifically mentioned in the articles, but, as I say, this isn't very clear in the manuscript. 4. In the current manuscript, there is no mention of what other studies have been done on content analysis around CCS in the news, and e.g. twitter etc. I would expect to see this in the introduction. 5. In the introduction there are some paragraphs that make many claims or statements with no references in them. Please provide references. E.g. page 3 line 6-12; page 5, lines 2-5, lines 6-12; page 6 line 1-14. 6. It is somewhat confusing to follow the discussions of timelines and events in Section 2.2, particularly on Page 9. Please add in dates to help the reader. For example, the year & common name of the Durban Climate Change Conference (COP17), when was Schwarze-Pumpe opened/closed and so on. In fact, I

would recommend compiling a timeline of CCS relevant events in Germany (and worldwide if relevant), and mapping the thematic analysis to this timeline. I would like to see plots like Fig 2 for e.g. each year analysed. Without this, there isn't evidence to back up the observations made. Also, a lot more could be said about the events/developments, for example, what was Schwarze-Pumpe? Why did BGR decide not to publish the report? What was the CCS law attempting to outline / achieve? Without more detail on these, it is difficult for the author or for the reader to map the thematic analysis results to the evolution of the CCS discussions in Germany. 7. Please define the themes depicted in Figure 2, either in the article main text or, better, in a table beneath the figure. What does "CCS exit" mean? What sort of thing is included in "pilot plant" theme, or "overview about CCS". 8. It is not very clear exactly what is determined to be "negative" or "neutral" or "positive" portrayal of CCS. Can the author give examples of what is looked for to determine this? I.e. the methodology / the issue. 9. The author presents table 1, of the degree of positive/negative articles for different actors. But what about for the different themes solicited in figure 2? Is there a trend for positivity for these different themes? I would expect, for example, "overview of CCS" to be neutral. 10. I would like to see results like those in Figure 2 and Table 1, for the four key stakeholder groups identified in the introduction. 11. In the paper there is a lot of discussion about the role of emotion in reporting, but a) little reflection on the importance of emotion as a communication tool and whether emotion itself is actually problematic (it seems to be presented as problematic in the article), and b) without any analysis of emotion in the content of the articles, as far as I can see. How might the author define 'emotional content' in the articles? Which have emotionally captivating headlines like that identified by Greenpeace in (page 5 line 20). 12. Page 12, line 4. Why would these conferences attract the attention of media? Many conferences don't have media coverage, because the public isn't really a key 'audience' for a conference, whereas for a political statement, the public is the audience. I find these sorts of statements a bit nonsensical. 13. I find the conclusions not only a bit jumbled and confusing with various grammatical errors, but also full of bold claims and subjective argument that seems to be uncoupled

from the study itself, and more towards an opinion piece. While I can see where the author is coming from, as I say, the statements are currently decoupled from the paper text. I strongly recommend revisiting the conclusions and re-writing them to be clearer, and to weave in the study outcomes much better.

TECHNICAL CORRECTIONS 1. The text refers to "we" and "the authors" but there is only one author. Should there be co-authors listed? If not, and this is truly has single authorship, then these need to be changed to be singular. 2. I find the current numbered system confusing. Section 2 is Analysis design, but then the results follow in subsections, leading me to think that these were also design elements rather than results. I suggest renumbering, with Section 2 = Methods; Section 3 = Analysis/results/discussion, section 4 = conclusions. 3. Figure 1 caption: should be May not Mai. Also the X axis units don't seem to be consistent. Please replot the figure with more comprehensible unit divisions. Also, see point 2 and 5 above) 4. Page 3, line 7: Please number the four stakeholder groups, else it looks like there are 5 or more here. 5. Page 12, line 12: "do surprise" should be "are surprising" 6. This is purely a preference thing, but I really don't like Public Interest Groups being called "PIGs", even though I know it is a commonly used acronym. This is because of the negative connotations that come with calling people pigs, and particularly when there is already tensions between dismissing public voices (like using phrases such as NIMBY and so on) especially around energy / politics. I suggest using simply IGs (Interest Groups) or community interest groups (CIGs - which although sounds like cigarettes is less offensive than pigs!) or even public or community interest groups (PCIGs) would be preferable. 7. Page 3 line 23 and the rest of that paragraph. H = hypothesis? If so, make this clear. 8. Page 5, line 23: "facts" can be a somewhat loaded word. Suggest avoid, and say "factual information" instead, or scientific information. 9. I don't think the author should assume that the reader knows where Brandenburg or Spremberg is, or that the Bundestag is the German Parliament, and so on. I suggest making all these things very clear, not only so that an international audience can follow the paper, but also so that the research outcomes are relevant to international audiences. 10. Likewise, I would always recommend defining ANY acronyms, no matter how 'obvious', so that the paper can be accessible to as many readers as possible. So, words like CCS, PR, NGOs, H should be explicitly made clear what they stand for. The author does this with some words (PIKS, BUND etc) but not all. 11. Page 9 line 6: Which four aspects? Please list them to make it explicit to the reader. 12. Page 11 line 7: by "energy providers" do you mean Vattenfall or do you mean all of them (EON, RWE, EnBW etc). 13. Table 1: caption /headings needs to be clearer. If these are actors, not subjects, then the categories are "Politicians" not politics, and e.g. "Scientists", "NGO representatives", "Economists".

---

## Referee Comment (RC2) · Anonymous Referee #2 · 20 Jul 2018

Interactive Comment on "The Takeover of Science Communication – Science Lost its Leading Role in the Public Discourse of Carbon Capture and Storage Research in Daily Newspapers in Germany" by Simon Schneider Anonymous Reviewer

Re: Research article "The Takeover of Science Communication – Science Lost its Leading Role in the Public Discourse of Carbon Capture and Storage Research in Daily Newspapers in Germany" by Simon Schneider.

General comments: The following article provides a content analysis of the German

newspaper coverage on CCS, using a quantitative approach for headline analysis as well as a qualitative content analysis of selected articles. The author's goal is to gain a better understanding of the role of science PR in the media coverage of CCS in Germany. Altogether, this approach promises to represent a substantial contribution to scientific progress within the scope of Geoscience Communication, as it is a suitable method to analyse/discuss the impact of science PR on media coverage. However, the current article reads more like an opinion paper and lacks an in-depth discussion and referencing in various sections. Therefore, in order to assure that the article will be relevant and insightful to an international and interdisciplinary readership, I suggest major revisions. I very much agree on the points raised by the reviewer Jen Roberts, and will only add some additional remarks.

Specific comments: 1. There are several sections where the author seems to describe a personal view without including appropriate references and without embedding his views into the wider discussion on the role of science PR within the media discourse on controversial energy technologies. For example, on page 3 the author writes "Research institutions and energy providers have tried to promote CCS as a transitional option to minimize the effects of climate change through the reduction of carbon dioxide emissions. (. . .) In contrast, the political arena in Germany has shown no great interest in contributing content and insight to the debate around CSS." Here, sound references that support this quite provocative claim would be helpful for the reader. Furthermore, the author outlines on page 5-6 in several sections that communication from CCS opponents is mainly using storytelling strategies and emotions to guide the audience's own reflections and interpretations. When the author writes on page 3 that "The allegation that CCS has been misused to improve a company's image can be found in the recurring argument that the otherwise climate-wrecking business activities of the energy providers are being "greenwashed" (Smid, 2009), and that "some NGOs nevertheless support research into and the development of CCS as a transitional measure", one might interpret that the author relates scientific integrity mainly to proponents of CCS. Similarly, it is not clear what the author exactly means by "harmful" (p.8,

[Figure]

Interactive
comment

line 19) or "negative" communication (page 11, line 11). Criteria for such categories would be needed to counter potential biases. Also, I find it problematic to describe four general stakeholder groups without differentiating the plurality of opinions within these groups. For example, the author writes on page 3 that "Research institutions and energy providers have tried to promote CCS as a transitional option to minimize the effects of climate change through the reduction of carbon dioxide emissions. Therefore, they seek to achieve public acceptance (...)" This can be perceived as misleading against the background that a variety of research institutions did not explicitly strive to achieve public acceptance for CCS, but rather wanted to provide the scientific basis for an open dialogue. On page 9-10, the author outlines that a lack of media resonance on the withheld study of the BGR (Federal Institute for Geo-Sciences and Raw Materials) shows that this study "was not considered to be of the utmost importance by journalists throughout Germany." Why doesn't the author further explore possible reasons for this lack of media coverage? The BGR is one of the most important consulting institutions of the German Federal Government but has also been subject to public criticism. This criticism was among others based on the fact that the energy company RWE financed staff positions in a research project of the BGR, which aimed to develop proposals for a binding set of rules for the use of CCS technology. I would appreciate if the author, given the controversial theme of interest-led science funding and its possible impact on scientific objectivity, more closely examines the theme of scientific independence. In the current version of the paper, one might think that the author sees science communication as entirely unbiased. Also, the author writes on page 12, that "it seems that the scientific field is unable to establish itself as a relevant source of information for journalists. Whether this is because of limited resources for outreach within scientific bodies or due to a misconception of how journalists seek out their sources remains unclear." I think this thought should be further explored. I cannot see sufficient evidence of the current study that journalists didn't consult scientifically reliable sources or the information from science PR. The main reason is that I do not understand which criteria the author used to choose the articles for his qualitative content analysis and how

the content analysis specifically took place. How does the author know that journalists didn't consult reliable scientific sources? In my opinion, the criteria that the author outlines (lack of mentioning of scientists, conferences and studies) doesn't necessarily mean that journalists didn't do proper research about the scientific basis of CCS. Furthermore, it would be good if the author could be more explicit about the scientific findings that he sees unrepresented. In the conclusion on page 14 (line 10) the author writes "If science wants to reestablish its position as a strong and constructive communication partner for journalists, science PR has to move toward a more intense deployment of emotion. Without such a change, journalistic communication will focus on actors from NGOs and PIGs because of the greater newsworthiness attributed to emotions and conflict." Regarding this finding, I find the way the author uses "emotion" as a general category very vague and risky, not only because it lacks a clear definition by the author, but also because it seems that the author suggests using "emotion" as a tool to create media attention. Here, I miss a critical reflection of the consequences of such an approach and also an exploration of other means for improving the communication/knowledge exchange between science PR and journalism.

Technical corrections: - Page 2, Line 24: focus instead of focuse

---

## Author Comment (AC1) · 2 Aug 2018

Dear Mrs Roberts, thanks a lot for your review. I particularly appreciate the depth and detailed comments and annotations. Since I am not familiar with the open peer review format, I will reply in more detail in a later stage of this publication process. While some of our comments are easy to answer and to integrate in the article, some annotations require further reflection from my side. Thanks again, Simon Schneider

---

## Author Comment (AC2) · 2 Aug 2018

Dear anonymous referee, thank you very much for your comments. I can understand your concerns and annotations and I am happy to get the chance to edit the article accordingly. As with the review from Mrs Roberts, most issues mentioned in your report have already been part of the study, but got somehow neglected in the writing process of the article. Therefore, please allow for thorough reflections before I will send a detailed response to your comments. Regards, Simon Schneider

---

## Author Comment (AC3) · 17 Aug 2018

Dear reviewer, thanks again for your comments on the article. Your remarks were very helpful to get some ideas and concepts clearer within the publication. Regarding your first comment, I have added a brief and attentive publication review of other studies about CCS in Germany to the article. This will help to understand the German CCS context in more detail.

The article invited to misinterpret the way scientific integrity might be related to CCS

proponents. This has been formulated in an unfortunate way. Within the revised version of the article, I tried to re-formulate this issue, so that scientific integrity now is related to responsible and thorough scientific process regardless of the attitude and outcome (pro or con) towards CCS.

The decision by BGR not to publish the map (this has actually been a more detailed register) of potential CCS storage sites cannot be evaluated from my perspective, since I have no insights into BGR decision making routines of that time. You are right, that those decision routines and the BGR's internal context would help to understand why the register was not published. Nevertheless, in regard to the analysis of media coverage of CCS, the BGR's internal motives in this respect are of no greater importance. This study focused more on selection routines of journalists and which actors were chosen as competent voices within the CCS theme.

Finally, your remarks about the use of the concept of "emotions" are well taken. While there are numerous concepts and theories about the role of emotions in communication, this study looks at emotions in quite a simplistic way. Nevertheless, I used the concept of emotions used by other social and communication scholars (e.g. Ungerer, 1997 or Wierzbicka,1995) before me. therefore, I have added some cross-references to studies, from which I have adopted the conceptualization of emotion.

Thanks again for your comments, that showed a great depth and understanding for the CCS theme and it's context in Germany.

Regards, Simon
* * *

---

## Author Comment (AC4) · 17 Aug 2018

Dear reviewer, thanks again for your comments on the article.

Your remarks were very helpful to get some ideas and concepts clearer within the publication. Regarding your first comment, I have added a brief and attentive publication review of other studies about CCS in Germany to the article. This will help to understand the German CCS context in more detail. To increase the readers' ability to understand the CCS context in Germany, I have added a timetable of relevant events

in Germany. You comment on the selection of articles by using the keyword coal are put into the German context. This will explain the validity of the used routine.

To illustrate the sampling process, I have added a schematic figure about the sampling routine. These additions will help to understand the analysis design and the methods used. Further corrections within the graphics used were made.

Your comment on the missing reflection of the concept of emotions has been taken seriously. I have, therefore, added some cross-references to scholars, from whom I have adopted the concept of emotions for this study. This will help to understand the used idea and concept of emotions more clearly. Thanks again for your comments on the article. I am sure, that your annotations are very helpful to raise the quality of this article.

Regards, Simon

---

## Referee Report (RR1)

This is a timely paper that presents some interesting findings on the agents involved in an important environmental science debate and the influence this plays on the nature of the media coverage. The conclusions presented at the end of the paper are supported by the data presented. However, conclusions would also be strengthened further by contrasting them with findings from the other literature that considers the role of science PR.

The social relevance of the study, of the need for actors from scientific groups to be part of science-related debates such as CCS, is presented convincingly in the introduction and this provides a firm basis for the study. However, there is a lack of clarity in some aspects of the introduction and in places assumptions are made that are not evidenced.

P4 (line 18) it states that scientific institutions do not include communication departments that follow equally high professional standards. What is meant by professional standards in this context needs to be defined as well as this claim being evidenced. As reported by Murcott and Williams (2012), numerous studies have noted the rise of science PR in universities, among other places, in recent years as well as the growth in effectiveness of these press offices at influencing media coverage. This is important context to the study.

P5 (lines 23 onwards) – how science communication is being defined is not clear. Journalistic representations of science are contrasted with the 'emotionless communication' behaviour of science. But where does this communication of science that's emotionless appear? And is journalism itself not part of science communication?

P6 (line 5) The material provided by science PR – focused on risks and benefits and demands and expectations – is contrasted with what recipients expect; factual and research based information. However, the preference for this material among audiences is not evidenced.

The analysis design is clearly explained and is robust, drawing on a commendable number of newspaper articles. However, the rationale behind using a quasi-week sum to reduce the number of articles could be clearer in terms of why it was important to eliminate single events; what are single events in this context and why were they irrelevant?

P9 – first paragraph, CCS is referred to as a scientific topic. However, what a scientific topic is and how that can be justified in relation to CSS is not clearly argued. While the legitimacy of analysing the actors present in media coverage and the influence of this on the nature of coverage cannot be questioned, the apparent claim to ownership of the topic by the scientific discipline can. So some justification is required for this.

P11 – The findings relating to the influence of the actors present in a story on whether CCS is framed negatively or positively are interesting.

P12 – (line 20) – The weakness of science PR in relation to other social systems is not a widely-held perspective in the literature. As stated above Murcott and Williams (2012) note the rise of science PR. The relative strength of science PR may be different in different countries. Studies noting the rise of science PR are worth reflecting upon in this paper. In anything, the contrast between the oft-reported rise of science PR and the findings here of the lack of a role for science in the CCS debate makes the findings more interesting.

P13 (line 19) 'failed conversation between researchers and scientists' – should this be journalists and scientists?

Finally the conclusion rightly mentions the frames used in the presentation of CCS in the media. Given the nature of the analysis conducted here, the paper would be strengthened by an earlier definition of framing theory and use of this during the analysis and results in addition to the use in the conclusion.

Murcott, T., Williams, A. (2012). The Challenges for Science Journalism in the UK. *Progress in Physical Geography*. 37 (2), pp. 152-160.

---

## Author Response (AR2)

Authors Response

Dear Editors,
Dear Reviewers,
thank you for helpful comments and suggestions. I incorporated most of your suggestions into the current version of the manuscript. Please find my point by point responses and changes made in the text outlined below.

With kind regards,
Simon Schneider
* * *
**Reviewer comment:** This is a timely paper that presents some interesting findings on the agents involved in an important environmental science debate and the influence this plays on the nature of the media coverage. The conclusions presented at the end of the paper are supported by the data presented. However, conclusions would also be strengthened further by contrasting them with findings from the other literature that considers the role of science PR.

The **Authors response:** Thank you for this comment. The ongoing debate about the role of science PR is of great importance – therefore, I have added a series of comments and references on this. In addition, earlier studies on the influence of science PR and the status of science communication from research institutions and organisations are referenced as well.

**Reviewer comment:** P4 (line 18) it states that scientific institutions do not include communication departments that follow equally high professional standards. What is meant by professional standards in this context needs to be defined as well as this claim being evidenced. As reported by Murcott and Williams (2012), numerous studies have noted the rise of science PR in universities, among other places, in recent years as well as the growth in effectiveness of these press offices at influencing media coverage. This is important context to the study.

**Authors response:** Thank you for this comment. Within the cited study by Höhn (2011), he observed some shortcomings in the professional development of science communication offices, according to 248 interviews universities and research organisations. Höhn sees structural deficits as pivotal for these shortcomings, which are obstacles for effective science communication. Höhn also mentioned, that the personal communication skills of communication and PR officers are predominantly excellent, but that for example missing acceptance and support from governance bodies as well as the lack of acknowledgement for outreach activities of scientists hinders a more intense communication process.

The recommended article by Murcott and Williams is a great example of external driving forces for science journalism, but does not look in detail on the level of proficiency in science PR at universities and research facilities. Nevertheless, Murcott and Williams give valuable insights into the context of science communication which should at least be mentioned when discussing the role of science in science communication. Therefore, I have added references to their article on two occasions in this publication.

**Reviewer comment:** P5 (lines 23 onwards) – how science communication is being defined is not clear. Journalistic representations of science are contrasted with the 'emotionless communication' behaviour of science. But where does this communication of science that's emotionless appear? And is journalism itself not part of science communication?

**Authors response:** Thanks for this suggestion. You are right in pointing out that the difference in science PR and science journalism is somewhat blurred in the beginning of this paragraph. Therefore, I added the terms at the appropriate positions within the text

to clarify what is meant. Science Communication includes both aspects as well as all other communication that is related to science.

**Reviewer comment:** P6 (line 5) The material provided by science PR – focused on risks and benefits and demands and expectations – is contrasted with what recipients expect; factual and research based information. However, the preference for this material among audiences is not evidenced.

> **Authors response:** Thank you for this comment. Empirical studies about the credibility of science journalism and science perception show, that the public (the lay-audiences) expect science to communicate in a factual way and from a neutral position. To support this statement, I have added a reference to an article by Maier et al, who analysed the expectations of different audience regarding the representation of science uncertainty. While Maier et al. focussed on the communication of scientific uncertainty, their findings also provide a more general perspective on science communication. As Maier states: 'They [the audiences] clearly expect fact-oriented information on the use of technology in everyday life.'

**Reviewer comment:** The analysis design is clearly explained and is robust, drawing on a commendable number of newspaper articles. However, the rationale behind using a quasi-week sum to reduce the number of articles could be clearer in terms of why it was important to eliminate single events; what are single events in this context and why were they irrelevant?

> **Authors response:** Thank you very much for this remark. I have added a short description of the motivation to apply the quasi-week sum approach which is used to reduce effects by for example thematic dossier series (a set of up to 10 articles about a single issue, mostly written by the same author or team of authors, published in a single outlet). These dossiers, which can be seen as singular (one day) publication maxima, would bias the analysis of the regional distribution of articles as well as of the temporal evolution of the topic. Therefore, it seems sensible to apply a method to minimize these effects, while not totally eliminate the dossiers from the analysis.

**Reviewer comment:** P9 – first paragraph, CCS is referred to as a scientific topic. However, what a scientific topic is and how that can be justified in relation to CSS is not clearly argued. While the legitimacy of analysing the actors present in media coverage and the influence of this on the nature of coverage cannot be questioned, the apparent claim to ownership of the topic by the scientific discipline can. So some justification is required for this.

> **Authors response:** Thank you for this critical remark. The suggested claim of ownership of the topic by science has to be explained more thoroughly. Therefore, I added a description on this by writing: 'Due to its technological and geo-chemical complexity as well as in respect to the focus on research driven feasibility studies such as Schwarze Pumpe, Ketzin and others, a more intense reference to relevant science was to be expected.' This includes the idea, that if a public discourse is intended by writing about CCS in a newspaper, CCS should at least be explained. Since the broader public cannot be expected to know the processes and details of CSS technologies, this explanation could have been made best from a scientific perspective. But only a small number of articles actually tried to explain what CCS is about. I think, that science has missed out a great chance for science communication in this respect. (My personal professional experience allows to say, that this is partly due to somewhat restrictive political guidelines for communication about CCS which were addressed to the coordination office GEOTECHNOLOGIEN who were responsible for at least parts of the CCS science communication efforts.)

**Reviewer comment:** P11 – The findings relating to the influence of the actors present in a story on whether CCS is framed negatively or positively are interesting.

> **Authors response:** Thanks a lot for this comment. I am thinking about publishing another article about this topic, too. I hope to do this in this journal later this year.

**Reviewer comment:** P12 – (line 20) – The weakness of science PR in relation to other social systems is not a widely-held perspective in the literature. As stated above Murcott and Williams (2012) note the rise of science PR. The relative strength of science PR may be different in different countries. Studies noting the rise of science PR are worth reflecting upon in this paper. In anything, the contrast between the oft- reported rise of science PR and the findings here of the lack of a role for science in the CCS debate makes the findings more interesting.

> **Authors response:** Thank you very much for this comment. Science communication at research institutes (science PR) is evaluated not only by how intense the media coverage of science itself is. Another relevant evaluation criteria is related to visibility of the institution itself, which is much easier to evaluate that the acceptance, an increase in legitimacy or the proposed increase in knowledge about science within the audiences. Therefore, it seems that science PR offices are more guided towards representation rather than towards starting a dialogue with society. This might not even be intentional, but demands for an internal debate about the services of science PR.
>
> While this issue cannot be address sufficiently within this article, I have added references to a number of studies on the role of science PR.

**Reviewer comment:** P13 (line 19) 'failed conversation between researchers and scientists' – should this be journalists and scientists?

> **Authors response:** Thanks for this correction. You are right – and I have changed 'researchers' into 'journalists'.

**Reviewer comment:** Finally, the conclusion rightly mentions the frames used in the presentation of CCS in the media. Given the nature of the analysis conducted here, the paper would be strengthened by an earlier definition of framing theory and use of this during the analysis and results in addition to the use in the conclusion.

> **Authors response:** Thank you very much for this recommendation. I have included some references and a short description of 'framing' in an earlier paragraph (3.2.) of the article. Since this article cannot discuss framing in an extent that would be sufficient for the ongoing debate within communication research, I have added a reference to an overview article, too.

**Additional remark:**
Since the Earth and Environmental Science at the University of Potsdam has undergone substantial structural changes, I have adjusted my affiliation accordingly.
I have added a short paragraph called Acknowledgement to the article.

[revised manuscript text omitted]